# The Effects of Phyllosphere Bacteria on Plant Physiology and Growth of Soybean Infected with *Pseudomonas syringae*

**DOI:** 10.3390/plants11192634

**Published:** 2022-10-07

**Authors:** Charles Agbavor, Babur S. Mirza, Alexander Wait

**Affiliations:** Department of Biology, Missouri State University, Springfield, MO 65897, USA

**Keywords:** phyllosphere bacteria, *Pseudomonas syringae pv. glycinea (Psg)*, gas exchange and growth of soybeans

## Abstract

Phyllosphere bacteria are an important determinant of plant growth and resistance to pathogens. However, the efficacy of phyllosphere bacteria in regulating infection of *Pseudomonas syringae pv. glycinea* (*Psg*) and its influence on soybean growth and physiology is unknown. In a greenhouse study, we assessed the influence of a phyllosphere bacterial consortium (BC) of 13 species isolated from field-grown soybean leaves on uninfected and deliberately *Psg* infected soybean plants. We measured *Psg* density on infected leaves with and without the application of the BC. The BC application resulted in a significant reduction in *Psg* cells. We also measured plant biomass, nodule mass and number, gas exchange, and leaf chlorophyll and nitrogen in four treatment groups: control plants, plants with a BC and no infection (BC), plants with BC and infected with *Psg* (BC + *Psg*), and plants infected with *Psg* alone. For all variables, plants infected with *Psg* alone showed significant reduction in measured variables compared to both BC treatments. Therefore, the bacterial consortium was effective in controlling the negative effects of *Psg* on growth and physiology. The BC treatment sometimes resulted in increases in measured variables such as plant biomass, nodule numbers, and leaf chlorophyll as compared to control and BC + *Psg* treatments. Overall, the positive influence of BC treatment on plant growth and physiology highlights its potential applications to increase crop yield and control bacterial pathogens.

## 1. Introduction

The phyllosphere is an important habitat for bacteria [1] that hosts a high abundance of diverse bacterial communities [2,3]. The phyllosphere surface area represents about 10^9^ km^2^ of the earth and is twice as big as the land surface area [4]. The number of bacterial cells on the leaves’ surfaces is suggested to be very high, up to 10^8^ cells/cm^2^ of leaf tissue [2,5]. Most of these phyllosphere bacteria are non-pathogenic and belong to three predominant bacterial phyla, including *Proteobacteria*, *Firmicutes*, and *Actinobacteria* [1,6,7,8]. Considering the high abundance of diverse bacterial populations on leaf surfaces of healthy plants suggests their potential role in plant growth and protection against plant diseases.

Several studies focused on the identification and classification of bacterial communities associated with the phyllosphere of soybeans [8,9,10] and reported a consistent presence of various bacterial genera such as *Pseudomonas*, *Sphingomonas*, *Methylobacterium*, *Streptomyces, Novosphingobium, Shigella* and *Acinetobacter*, *Bacillus, Arthrobacter*, and *Pantoea* as a part of phyllosphere bacteria [3,8,10,11,12,13]. Some of the suggested mechanisms through which these epiphytic phyllosphere bacteria improve growth of the host plant include phytohormone production [14,15], controlling the growth of plant pathogens [12,16] and helping the host plant to tolerate high salt concentrations [17]. So far, limited information is available on the potential influence of phyllosphere bacterial application on plant physiology and growth variables such as photosynthesis, chlorophyll concentrations, nitrogen content of leaves, mass and number of root nodules, and plant growth in response to *Psg* infection.

In general, most phyllosphere bacteria are beneficial to plants [6,18,19]. However, some bacterial genera are considered plant pathogens or opportunistic pathogens that can infect a wide spectrum of host plants [20]. Many different pathogenic species have been identified that can infect economically valuable crops [21]. *Pseudomonas syringae* is a model bacterial pathogen that is frequently used for studying host–pathogen interactions [22,23]. *Pseudomonas syringae pv. glycinea* causes bacterial blight disease in soybean, which is a common disease of soybean and causes average soybean yield losses of 11% annually in the United States [24,25,26,27]. One of the suggested mechanisms through which host plants can potentially resist these bacterial infections is through the presence of other bacteria on leaves [16,28]. For example, a study by Berg and Koskella [28] observed that a microbiome-based application of commensal bacteria protected tomato plants against *P. syringae pathovar tomato*. However, limited information is available on the potential application of phyllosphere bacteria to improve plant growth and control bacterial diseases. Therefore, the focus of the current study was to assess the potential influence of host bacterial species’ application on the survival of *Psg* and its effects on soybean physiology and growth.

The main objectives of the current study were to assess: (i) the potential role of bacterial consortium application in controlling *Psg* infections, and (ii) the potential influence of the application of a bacterial consortium obtained from healthy soybean plants on the growth and physiology of soybean plants with and without *Psg* infection.

## 2. Results

### 2.1. Bacterial Consortium

As a representative of phyllosphere bacteria, we isolated 26 bacterial strains based on their cell shape, colony morphology and the growth rate on two different growth media. These isolates were further identified by 16S rRNA gene sequencing which suggests the presence of 13 unique strains. These isolates were grouped into three different bacterial phyla based on the 16S rRNA gene sequencing; *Proteobacteria*, *Firmicutes*, and *Actinobacteria*. A majority of isolates (42%) belonged to *Proteobacteria*, followed by *Firmicutes* (38%) and *Actinobacteria* (19%). Within the phylum *Proteobacteria,* we identified five unique strains that were related to *Pseudomonas*, *Pantoea*, *Brevundimonas*, and *Roseomonas* species. The five unique isolates of *Firmicutes* phyla were related to the *Bacillus* and *Paenibacillus* species. The three unique *Actinobacteria* isolates were related to *Curtobacterium*, *Cellulomonas*, and *Microbacterium* species (Figure 1). These 13 bacterial species obtained from leaves of field-grown soybean plants were used to make a representative BC (Figure 1).

### 2.2. Influence of Bacterial Consortium on the Growth or Survival of P. syringae Cells on Leaves

The application of the BC reduced the number of *Psg* cells per leaf disc by a factor of two compared to no BC-treated *Psg* treatment (Figure 2). Therefore, there is evidence that the BC negatively affected the growth of *Psg.*

### 2.3. Influence of Bacterial Consortium on Plant Growth

The BC application resulted in the significant influence of different plant growth parameters (Figure 3A–D). For example, BC application to uninfected plants significantly increased plant shoot dry weight compared to the control (Figure 3A). In addition, the shoot dry weight of the BC + *Psg* treated group was significantly higher than the *Psg* treated group, suggesting a positive effect of the BC in controlling *Psg* infection (Figure 3A). Comparing across treatments and greenhouse bays, the shoot dry weight of the *Psg* treated plants was also significantly lower than the uninoculated control (Figure 3A). However, the shoot dry weight of the control and the BC + *Psg* treated group was not significantly different, indicating that the BC does not completely ameliorate the negative effects of *Psg* (Figure 3A). 

The BC treatment added to uninfected plants had a significant positive effect on nodule fresh weight as compared to the control treatment (Figure 3C). The dry shoot weight, fresh nodule weight, and number of nodules in the BC + *Psg* treatment was not significantly different from the control (Figure 3A,C,D). The dry shoot weight, fresh nodule weight, and number of nodules were significantly lower in the *Psg* treatment than all other treatments (Figure 3A,C,D), thus confirming a negative effect of the pathogen. 

### 2.4. Influence of Bacterial Consortium on Leaf N and Chlorophyll Content

The nitrogen content (%) of the BC-treated plants with and without *Psg* were significantly higher than the *Psg* treatment (Figure 4A). Additionally, we observed that leaves in the BC + *Psg* showed a small number of yellowing and brown spots compared to *Psg* treatment alone (Appendix A). These results again confirm the negative effect of *Psg*, and the positive effects of a BC. 

Leaf chlorophyll concentrations were significantly higher in the BC treatment added to uninfected plants compared to the control, indicating a positive effect even in the absence of *Psg* (Figure 4B). The chlorophyll concentrations of the control and the BC + *Psg* treatment were significantly different, and this was also true for the control and the *Psg* treatment (Figure 4B). Therefore, the BC can protect the host plant from *Psg* infection.

### 2.5. Influence of Bacterial Consortium on Plant Physiology

The photosynthetic potential measured as photosaturated photosynthetic rates (A_max_) of BC added to uninfected plants was not significantly different than rates in control plants (Figure 5A). However, BC added to infected plants resulted in significantly higher photosynthetic potential, illustrating the positive effects of the BC (Figure 5A). Net photosynthetic rates at growth photosynthetically active radiation (A_growth_) followed the same pattern (Figure 5B). The net photosynthetic rate of the *Psg* treatment was significantly lower as compared to the BC + *Psg* and control treatment (Figure 5B). The net photosynthetic rates of BC + *Psg* treated plants was significantly higher compared to the *Psg* treated plants (Figure 5B).

The uninfected plants treated with BC had a significantly higher rate of transpiration than control plants (Figure 5C), which is not entirely consistent with photosynthetic rates, but is consistent with dry mass accumulation (see Figure 3A). *Psg* treated plants had a significantly lower transpiration rate compared to the control and *Psg* plants treated with the BC (Figure 5C). Therefore, the BC was effective in protecting the host plant from the negative effects of *Psg* with respect to the tradeoff between carbon gain and water loss. However, instantaneous water use efficiency across different treatments followed a slightly different pattern, with the BC added to uninfected plants being significantly lower than control, and BC inoculated plants not being significantly different than *Psg* treated plants (Figure 5D).

## 3. Discussion

Phyllosphere bacteria have been shown to be an important determinant of plant growth and resistance to microbial infections [29,30,31]. An applied microbial consortium has been shown to protect a host plant against plant pathogens [31]. The two-fold reduction in *Psg* cells recorded from leaf discs in this study illustrates protection provided by the application of bacterial consortium, which results in robust growth (Figure 3A–D). 

The beneficial phyllosphere bacteria can protect the host plants against pathogens, by directly competing for nutrients and space [11], interfering with their communication [32], excreting antimicrobial metabolites or enzymes [33], and/or parasitizing pathogens. Alternatively, these beneficial microbes can trigger the plant’s immune response and modulate plant hormone levels to inhibit growth and establishment of pathogens [34,35]. 

Overall, we observed a significant effect of the BC application in controlling *Psg* infection. Most of the isolates used in the BC (Figure 1) have been previously reported as a part of the phyllosphere microbiome of various crops and have been suggested to contribute positively toward plant growth. For example, within Proteobacteria, five bacterial isolates used in the BC were related to *Pantoea*, *Pseudomonas*, *Roseomonas*, and two *Brevundimonas* strains. *Pantoea* species were reported to produce antibiotics such as pantocins [36], herbicolins [37], and phenazines [38] which inhibited the growth of pathogens like *E. amylovora* and *X. campestris* species [35,39,40,41,42]. Some *Pantoea* species were reported to fix atmospheric nitrogen [43], produce plant-growth hormones [44], and N-acyl-homoserine lactone (AHL) molecules to suppress pathogens’ growth on leaves [42]. Similarly, a previous study by [45] reported an application of *Brevundimonas* species along with the *Microbacterium* species suppressed the blight disease of *Anthurium* species and provided protection against *Xanthomonas* infections. Likewise, *Pseudomonas* species also previously demonstrated biocontrol activity on leaves producing phenazines [46], siderophores [47], 4-hydroxy-2-alkylquinolines [48], volatile compounds such as cyanide [49], and cyclic lipopeptides [50], and rhamnolipids [48,51,52]. Some of the *Pseudomonas* species were also reported to produce Rhs proteins (rearrangement hot spot) through VI secretion system to inhibit *P. syringae* and other pathogenic infections [53]. *Pseudomonas* species such as *P. chlororaphis* (MA342), *P. putida*, *P. koreensis*, and *Pseudomonas* sp. DSMZ 13134 were also used as a commercial biocontrol agent to reduce the growth of plant pathogens [11,54]. Although most of the *Pseudomonas* species are reported as plant beneficial microbes, *Psg* is a well-known plant pathogen.

Like Proteobacteria, within Actinobacteria three bacterial isolates used in BC were related to *Microbacterium*, *Curtobacterium*, and *Cellulomonas* genera. Previously, *Microbacterium* species were detected on spinach and lettuce phyllosphere [55,56] and showed antagonistic activity against a pathogenic *E. coli* strain [56] and fungal plant pathogens [57]. *Microbacterium* species were also suggested to play a role as a biocontrol agent by degrading signaling molecules (AHLs) produced by plant pathogens in the phyllosphere [58]. Members of the *Curtobacterium* genus are commonly found in the phyllosphere of various plants [59,60] and are suggested to perform important ecological roles such as protection against disease [61] and increasing plant growth [62]. However, one of the species within the genus *Curtobacterium* (*C. flaccumfaciens*) is considered to have a negative effect on soybean growth [63]. The isolation of *Curtobacterium* and *Cellulomonas* species has been reported from potato phyllosphere grown in a greenhouse [64]. Similarly, the positive influence of other strains used in BC in controlling plant pathogens were also observed previously.

So far, less is known about the potential influence of phyllosphere bacteria on physiological activities that are related to plant growth. We observed a positive influence of BC inoculation on the growth and physiological activities of the BC-treated plants. For example, higher nodule weight and number of nodules in the BC treated group (Figure 3C,D) suggest either a direct or an indirect effect of the phyllosphere bacteria on nodulation. The lower number of nodules in *Psg* treatment could be due to the host plant’s response to infection. Previously, leguminous plants such as *L. japonicus* and *M truncatula* [65] were suggested to produce jasmonic acids and innate immune effector proteins in response to infection which causes down-regulation of nodules formation. Furthermore, pathogens like *Pseudomonas syringae pv.* tomato strain was suggested to inhibit the expression of genes associated with nodule formation in *M. truncatula* [66]. 

We observed a higher amount of nitrogen in the BC-treated groups (Figure 4A) which could be due to the presence of nitrogen-fixing bacteria such as *Paenibacillus* sp. [67] in BC. The reduced amount of assimilated nitrogen in the leaves of the *Psg* treatment (Figure 4A) may be due to production of the phytotoxin coronatine and coronamic acid by *Psg* strain [68] or due to the indirect effect of *Psg* on nodulation and rate of nitrogen fixation as mentioned above. The chlorophyll content in the BC-inoculated plants were also higher (Figure 4B). Chlorophyll is a light harvesting molecule, and its concentration is related to the photosynthetic efficiency of the plant [69]. The increased chlorophyll concentrations, therefore, suggest a high impact on the overall photosynthetic rate, which was observed in BC-treated plants (Figure 5A,B). 

We observed lower photosynthetic rates in the *Psg* treated plants (Figure 5A,B) which could be due to leaf nitrogen and chlorophyll and/or biotic stress caused by *Psg* infection [70,71]. The *Psg* causes the formation of chlorotic halos on the leaves which reduces the light harvesting by chlorophyll [72]. Previously, *Pseudomonas syringae pv. tabaci* has been shown to decrease the net photosynthetic rate in tobacco leaves [73]. The results from this study also suggest faster CO_2_ assimilation owing to rapid ATP and NADPH generation in the BC-treated plants (higher A_max_, see Figure 5A).

Transpiration helps in the transport of water and nutrients in plants [72]. Pathogens can reduce transpiration with concomitant decreases in carbon gain or direct water stress in leaves. This is because pathogens can modulate the opening and closing of the stomata directly or indirectly [74,75,76]. Conversely, the phyllosphere bacteria may have the potential for improving the efficiency of transpiration in plants. 

Water use efficiency indicates the amount of water lost to the amount of carbon fixed [77] and is important in understanding water management problems and productivity in agriculture [78,79]. The results of this study suggest the potential of increasing carbon gain without a corresponding increase in transpiration in plants by the inoculation of the phyllosphere bacteria (Figure 5D). Notwithstanding, the role of phyllosphere bacteria in improving water use in plants is unexplored and the development of new approaches in the control of excessive water loss will be helpful in areas where there is drought and high salinity [80]. While photosynthetic rates were higher in BC treatments (Figure 5B), this did not result in an expected tradeoff with water use efficiency (Figure 5D). Increasing crop yield without decreasing water use efficiency is a major goal of agricultural research and has been linked to bacteria [81,82].

## 4. Materials and Methods

### 4.1. Experimental Overview

Soybean (*Glycine max* L. Merrill.) seeds were grown in two climate-controlled greenhouses at Missouri State University, Springfield, Missouri (37.1981° N, 93.2815° W). Soybean seeds (AG49X6) were purchased from Associated Seed Growers, Inc. (Asgrow, MI, USA). Seeds were disinfected and inoculated with *Bradyrhizobium japonicum* (details are described in the following sections). The germinated seedlings were transplanted in a sterilized growing mix of pH 6.8. Plants were categorized into four treatment groups: Control, bacterial consortium (BC), bacterial consortium + *P. syringae* infection (BC + *Psg*), and *P. syringae* infection (*Psg*). Two of four treatments were sprayed with or without *Pseudomonas syringae pv. glycinea* (2 × 10^7^ cells/mL) and the BC inoculum. We quantified the number of *Psg* cells on soybean leaves after day 20, 35 and 45 of plant growth. Plant biomass accumulation, leaf nitrogen and chlorophyll content and gas exchange rates were measured after 20, 35 and 45 days of plant growth. We reported the data for day 35 only but the overall trend of differences across various treatments was similar on day 20 and day 45 of plant growth (see Appendix A).

### 4.2. Isolation of Bacterial Species and Development of Bacterial Consortium

We collected soybean leaves from twenty healthy plants growing on separate farms (Kindrick farms managed by the William H. Darr College of Agriculture and another farm in Buffalo area, Missouri). The plants were at the V4–V5 plant growth stage, which is vegetative growth of soybean producing 4th and 5th unfolded trifoliolate leaves. The leaves were transported to the laboratory on ice. Three leaves per plant were gently rinsed in 10 mM MgCl_2_ buffer to wash off the soil particles. Three cm leaf discs (2 discs per leaf) were cut using a sterile cork-borer. The leaf discs were pooled (total 120 discs), transferred into 500 mL conical flasks containing 10 mM MgCl_2_ buffer and stirred for 20 min using the magnetic stirrer to wash out epiphytic bacteria from leaf discs into the buffer. The solution was further centrifuged at 12,000 rpm for 5 min to concentrate and harvest bacterial cells. Serial dilutions up to 10^−4^ were prepared and 100 µL aliquots from the dilutions were transferred to sterile tryptic soy agar (TSA) and nutrient agar (three replicates per dilution) plates. The plates were incubated at 30 °C for 2–7 days. To estimate the approximate number of bacterial colonies under natural field conditions, we counted these colonies using a manual colony counter (Leica Quebec Darkfield Colony Counter). The bacteria cell density per cm^2^ of leaf area was calculated using the formula: Bacteria cm^−2^ = Total no. of colonies per µL/Total area of 60 leaf discs × 2. The area of one leaf disc = πr^2^ (where r is the radius of a disc in cm). This bacterial cell density information from field-grown soybean leaves was used as a guide to determine the number of cells applied in the bacterial consortium treatment. The bacterial isolates were purified by streaking on agar plates (TSA and nutrient agar) and pure cultures were isolated. The isolation protocol was adopted and modified from two published sources [28,83]. A total of 26 bacterial isolates based on different colony morphologies (size, shape, margins, color, etc.) or growth rates were selected and identified using 16S rRNA gene sequencing. Based on 16S rRNA gene sequencing, we identified 13 unique bacterial strains that were used to develop multi-species bacterial consortium (BC). The BC was sprayed on the soybean plants after 12 and 24 days of plant growth and *Psg* cells (2 × 10^7^ cells/mL) were applied on day 16 and 31. Bacterial consortium was developed by growing each culture individually to the same cell density by measuring OD_600_ and one ml of individual cultures were pooled together in MgCl_2_ buffer of 500 mL in a sterile spray bottle for spraying the soybean leaves. The final concentrations of each species in the consortia was approximately 1500 CFUs/mL.

### 4.3. Microbial DNA Extraction and 16S rRNA Amplification 

Bacterial DNA from pure cultures were extracted using the DNeasy PowerLyzer Kit (Qiagen, Hilden, Germany) following the manufacturer’s protocol. The purity and concentration of the DNA samples were measured with the nanophotometer (Implen, Westlake Village, CA, USA) and the samples were stored at −20 °C. The 16S rRNA gene from the bacteria were amplified with the 8F (5′-AGAGTTTGATCCTGGCTCAG-3′) and 1492R (5′-TACCTTGTTACGACTT-3′) primers in a reaction volume of 25 μL containing 0.2 mM dNTPS, 10 ng/μL of the forward and reverse primers, 1 × PCR buffer, 2 mM MgSO_4_, 1 μL of template DNA, and 2 U of *Taq* DNA polymerase (GenScript, Piscataway, NJ, USA). The PCR conditions were: 35 cycles of temperature cycling of 95 °C for 45 s, 56 °C for 30 s, and 72 °C for 45 s, and a final extension at 72 °C for 7 min. The resulting amplicons were visualized by electrophoresis using 1% agarose gel stained with Ethidium Bromide. The PCR product was cleaned with ExoSAP-IT (USB, Cleveland, OH, USA) and bidirectional sequencing was performed with the 8F and the 1492R primers using the 3730XL Genetic Analyzer (Life Technologies, Applied Biosystems).

### 4.4. Bacterial Identification and Phylogenetic Analysis

Bacterial 16S rRNA sequences from 26 isolates were edited using the Sequencher (version 4.5.6; Gene Codes Corp., Ann Arbor, MI, USA). A contiguous sequence was generated for each isolate that was used for bacterial identification using the National Centre for Biotechnology Information blast search tool. Details were the same as those described previously [84].

Bacterial 16S rRNA gene sequences from different isolates were aligned and used for creating a maximum likelihood (ML)-based phylogenetic analysis. The MEGA software version 5.2 [85] was used to construct a ML phylogenetic tree. The detailed settings for the phylogenetic analysis were the same as those described previously [86]. The confidence in tree topology was assessed by 10,000 bootstrap iterations. The bootstrap values greater than 50% were reported in the phylogenetic tree. The closely related sequences from GenBank that were related to our isolates are also reported (Figure 1).

### 4.5. Pathogen Culture

*Pseudomonas syringae pathovar glycinea* (ATCC^®^ 8727™) culture was obtained from American Type Culture Collection (ATCC) and grown on nutrient agar media at 30 °C for 24 h [12,87]. Colony forming units (CFU) and optical density at 600 nm (OD_600_) were measured using BioMate™ 3S Spectrophotometer (ThermoFisher Scientific, Waltham, MA, USA). 

### 4.6. Sterilizing Seeds and Planting

The sterilization of soybean seeds was performed as previously described [88]. Briefly, healthy soybean seeds were placed in 15 mL falcon^®^ tubes. In a 500 mL graduated beaker, 100 mL of sodium hypochlorite solution (6%) was added and placed in a desiccator. While under the desiccator, 3 mL of concentrated hydrochloric acid was added dropwise to the beaker containing bleach. The soybean seeds were kept under the hood for 3 h to ensure sterilization by the vapor generated. The sterilized soybean seeds were rinsed with sterile water, soaked in 0.8% of water agar for a week and the seedlings were transplanted (one seedling per pot) into free-draining pots (25 cm diameter by 40 cm height) containing potting media (Jolly Garden, Pro-Line C/20. Growing mix of Canadian sphagnum peat and perlite) was filled ¾. Soybean plants were maintained in the greenhouse at 26–28 °C, 50–70% relative humidity, and 14 h light/10 h dark each day (photosynthetically active radiation > 700 umol m^−2^ s^−1^). Two treatments were grown in one greenhouse bay (Control, BC) and two treatments in another (a door was the only connection between bays) (BC + *Psg* and *Psg*). This separation was done to prevent the aerial transmittance of *Psg* cells to control and BC treatments. Limiting treatments to different greenhouse bays allowed us to test the effects of the BC on plants that were not infected with *Psg* cells. While this is a potential limitation in making comparisons across all four treatments, we believe the pairwise comparisons of means illustrate how the consortium might affect soybeans that are uninfected or are infected with *Psg*. For a presentation of treatment effects, we ordered them as Control, BC, BC + *Psg*, *Psg.* Plastic pots were positioned such that a space of 45 cm existed between them on the bench. The pots were completely randomized every week to reduce confounding factors due to microenvironment in greenhouse bays and to ensure uniform growth. 

### 4.7. Pseudomonas Syringae Pathovar Glycinea Quantification

The number of *Psg* cells per leaf disc was quantified for twenty plants per treatment. Three leaves per plant were sampled and six leaf discs per plant were generated. *Psg* cells per leaf disc were quantified by vigorous shaking of discs in 2 mL of 10 mM MgCl_2_. Serial dilutions of leaf discs washed material was plated on the nutrient agar plates and *Psg* colonies were counted. The *Psg* cells could be distinguished based on the specific *Psg* colony morphology and faster growth rate as compared to other cells in BC. The *Psg* colonies appeared approximately 4–5 h before other BC cells. We compared *Psg* colonies from the leaf discs along with the control plates containing only pure culture of *Psg* cells for growth rate and colony morphology. 

### 4.8. Plant Growth

Soybean shoots and roots were harvested (20 plants per treatment for each of the three sampling points), and the fresh weights were determined to the nearest 0.01 g. The shoot samples were dried in a Quincy Lab 40AF-1 Forced Air oven (Quincy Labs, Chicago, IL, USA) for 24–48 h at 100 °C and the dry weights were determined using the Ohaus Explorer Analytical balance (Ohaus, NJ, USA). Root nodules from 10 of 20 randomly selected plants per treatment were carefully separated, counted and weighed using the Ohaus Explorer Analytical balance (Ohaus, NJ, USA).

### 4.9. Measurement of Chlorophyll

Chlorophyll concentrations were measured with an MC-100 Chlorophyll Concentration Meter (Apogee Instruments, Inc., Logan, UT, USA). Measurements were done by clipping the sensor onto the leaf at three distinct points and the results were averaged. Measurements from 50 plants per treatment per sampling day were taken between the margin and the mid-rib of the leaf [89]. An area of 63.6 mm^2^ (9 mm diameter), 19.6 mm² (5 mm diameter with reducer) of the leaves was measured by the instrument and the results were displayed as µmol of chlorophyll per m^2^ of leaf surface. The MC-100 measures the relative chlorophyll concentrations by measuring the absorbance at 653 nm and 931 nm. The chlorophyll concentration index was calculated by dividing the percent transmittance (T) at 931 nm by T at 653 nm and the meter was internally calibrated for soybean.

### 4.10. Gas Exchange

Photosynthetic rates (μmol CO_2_ m^−2^ s^−1^), transpiration rates (mol H_2_O m^−2^ s^−1^), and stomatal conductance to H_2_O (mol H_2_O m^−2^ s^−1^) were measured using a LI-6400XT portable gas exchange system (LI-COR Biosciences, Lincoln, NE, USA). Water use efficiency was calculated as the ratio of the net photosynthesis divided by the transpiration rate. The IRGA leaf chamber was set to a temperature of 24 °C and the partial pressure of carbon dioxide was set at a reference point of 400 μmol CO_2_/mol. The relative humidity was controlled at approximately 60% in the chamber to equal that in the greenhouse. All gas exchange measurements were taken between 11 am to 2 pm and microclimatic conditions in the greenhouse were set to match conditions in the leaf chamber. Leaf area for measurements was 6 cm^2^ and the midrib of the plants were centered in the leaf chamber.

### 4.11. Percent Nitrogen

Leaf disc samples were collected, homogenized using a Bullet Blender Tissue Homogenizer (Next Advance, Troy, NY, USA), weighed and analyzed using the EA-Isolink elemental analyzer interfaced to a Delta V advantage (Thermo Electron, Bremen, Germany). The leaf disc samples were combusted in the presence of a small amount of oxygen in a helium stream causing nitro oxides to be reduced to nitrogen. The nitrogen gases were separated on a 0.5-m N column. The resulting gases were admitted into the mass spectrometer via a ConFlo IV along with reference gas pulses. Data were normalized using a two-point isotopic correction of standards and N (%) were calculated from a standard calibration curve. The leaf N analysis was carried out at the University of Arkansas in a Stable Isotope Lab.

### 4.12. Statistical Analysis

All data were analyzed using Minitab software version 19 (Minitab, LLC, State College, PA, USA). The effects of the bacterial consortium on the *Pseudomonas syringae pv. glycinea* were analyzed using a one-way ANOVA. Tukey’s pairwise comparison was used to assess all pairwise comparisons. The light response and CO_2_ response curves (Aci) curves were modeled using a modified R package for gas exchange variables [90]. All conclusions are based on the type-I error rate of 0.05.

## 5. Conclusions

In summary, the result of our study suggests that bacterial consortium application can significantly improve plant growth and provides protection to the host plants against *Psg* infection. The members of BC can be a potential alternative to bactericide applications which can have a detrimental effect on other plant-beneficial bacteria. Future studies will focus on identifying the potential mechanisms through which members of BC enhance plant growth and provides protection to the host plant against *Psg* infections. 

## Figures and Tables

**Figure 1 plants-11-02634-f001:**
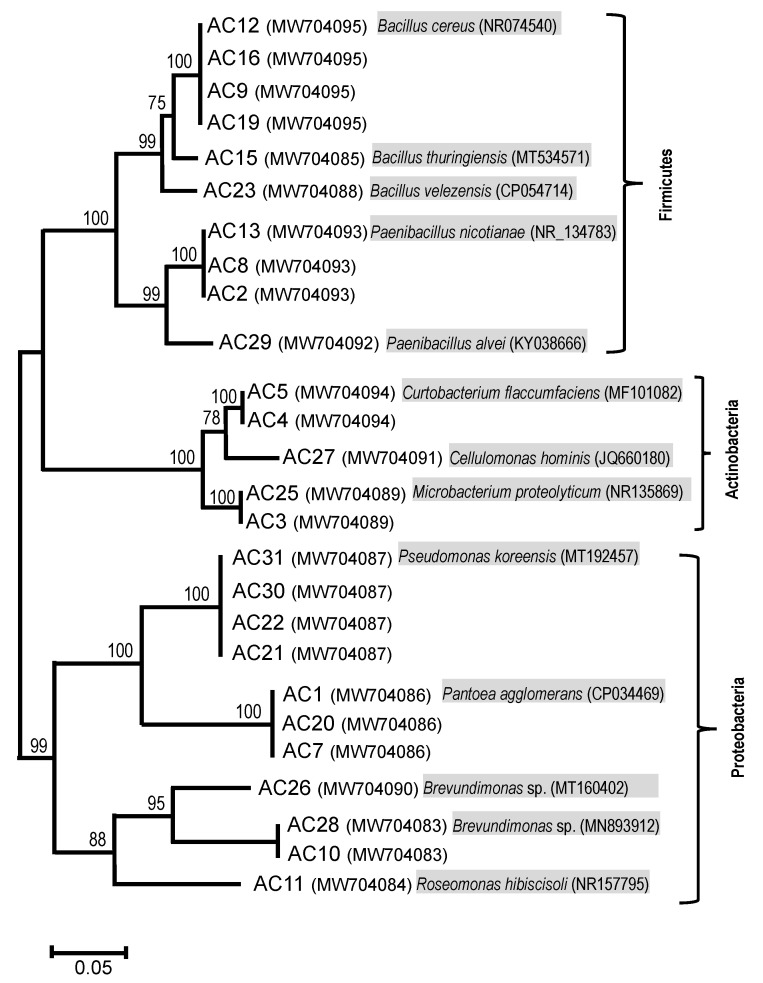
Maximum likelihood phylogenetic tree based on partial sequences of the 16S rRNA gene of the 26 isolates from the leaves of field-grown soybean leaves. We used 13 unique species to create a bacterial consortium. Numbers above the nodes represent the maximum likelihood of bootstrap support above 70%. Numbers in the parenthesis next to the name of the isolate represent the GenBank accession number. The most closely related organism identified based on the NCBI blast search were also reported next to the name of the isolated strain (shaded in gray). Clustering at the phylum level was shown on the right.

**Figure 2 plants-11-02634-f002:**
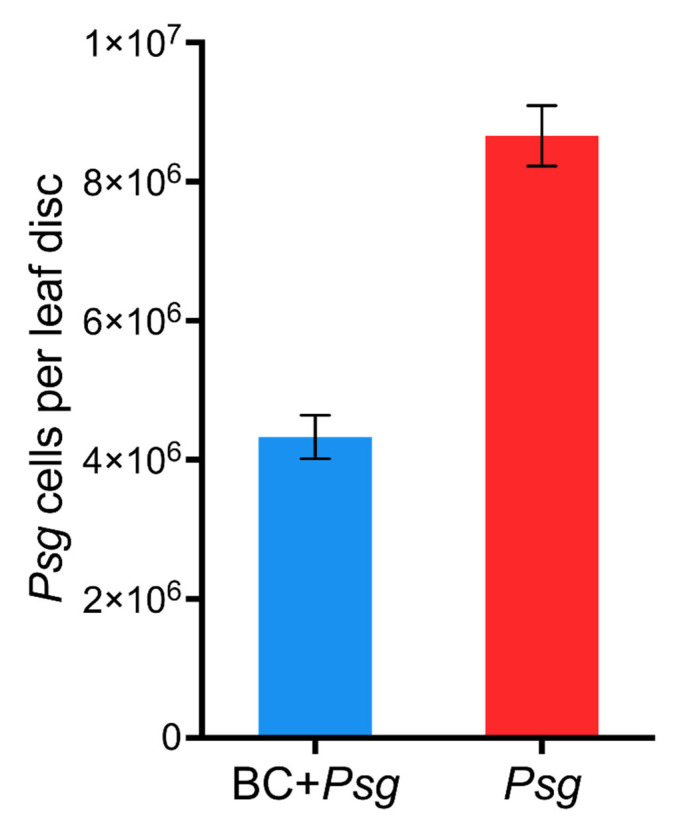
Quantification of *Pseudomonas syringae pv. glycinea* (*Psg*) cells per leaf disc in *Psg* and BC + *Psg* treatments on day 35 of plant growth. BC (bacterial consortia). The number of *Psg* cells per leaf disc was determined for twenty plants per treatment. Three leaves per plant was sampled and six leaf discs per plant were generated. The amount of *Psg* cells per leaf disc was significantly higher in the *Psg* treatment alone compared to the BC+ *Psg* treatment (F = 6.40; *p* < 0.016).

**Figure 3 plants-11-02634-f003:**
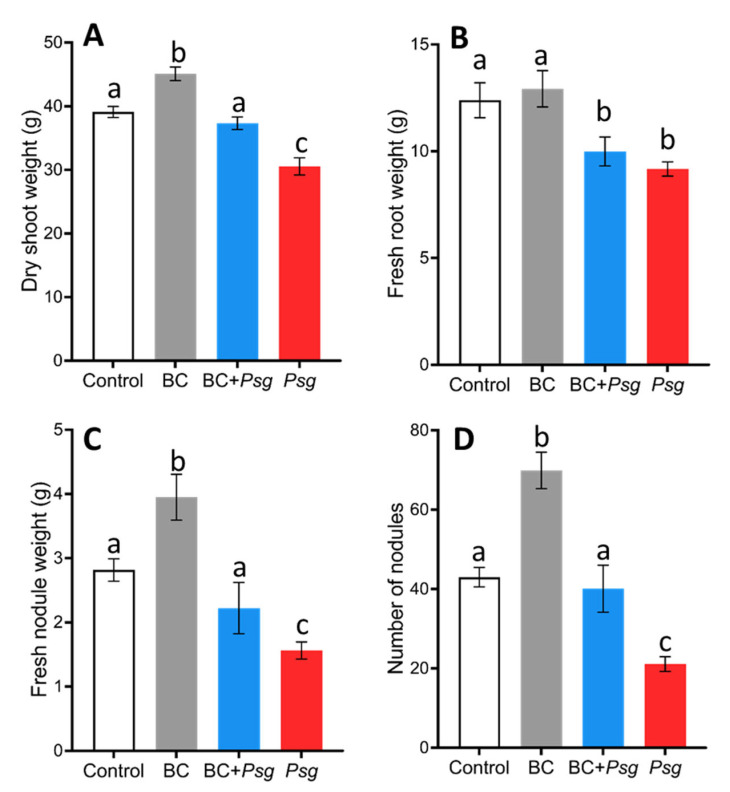
(**A**–**D**). Variations in mean soybean plant growth parameters, including dry shoot weight (**A**), fresh root weight (**B**), fresh nodule weight (**C**), and the number of nodules (**D**) across four different treatment groups on day 35 of plant growth. Treatment had a significant effect on dry shoot (F = 30.77; *p* < 0.0005) and fresh root weights (F = 7.4; *p* < 0.0005). For both dry shoot and fresh root weight 20 plants per treatment were measured. The nodule weight (F = 12.30; *p* < 0.0005) and the number of nodules (F = 24.95; *p* < 0.0005) per plant were also different. We collected 10 root nodules per plant from 10 randomly selected plants for each treatment and were analyzed for fresh nodule weight. The treatments that do not share the same letter are significantly different (alpha = 0.05, Tukey-adjusted). Error bars represent the standard error of the mean. Bacterial consortium (BC).

**Figure 4 plants-11-02634-f004:**
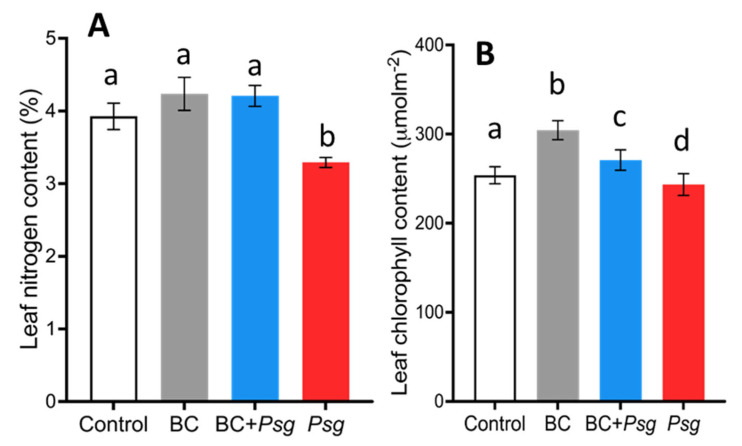
(**A**,**B**). Variations in mean leaf nitrogen content (**A**) and chlorophyll concentration (**B**) across four treatment groups on day 35 of plant growth. Treatments had a significant effect on the nitrogen content (F = 3.59; *p* < 0.034; sample size 10 plants per treatment) and chlorophyll concentration (F = 47.73; *p* < 0.005; sample size 50 plants per treatment). For every plant, the chlorophyll concentration of three leaves was measured and averaged. Three distinct areas on every leaf were sampled. The treatments that do not share the same letter are significantly different (alpha = 0.05, Tukey-adjusted).

**Figure 5 plants-11-02634-f005:**
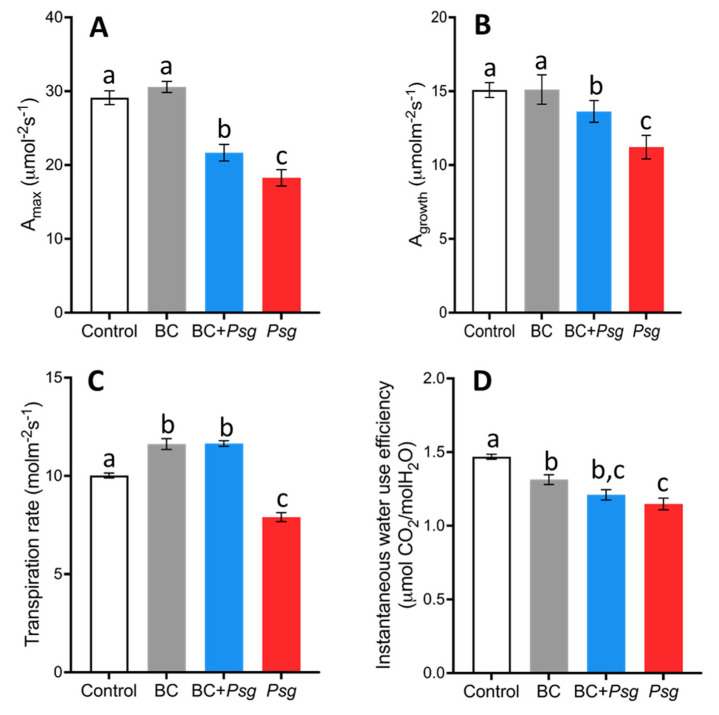
(**A**–**D**). Variations in mean photosaturated photosynthetic rate A_max_ (**A**), photosynthetic rate at growth photosynthetically active radiation levels (**B**), transpiration rate (**C**) and instantaneous water use efficiency (**D**) across four different treatment groups on day 35. Treatment had a significant effect on A_max_ (F = 35.0; *p* < 0.0005), A_growth_ (F = 92.95; *p* < 0.0005), transpiration rate (F = 76.0; *p* < 0.0005) and water use efficiency (F = 20.64; *p* < 0.0005) on day 35. The sample size for these variables was 10 plants per treatment. All parameters were measured at ambient PAR levels in the greenhouse as mentioned in Section 4. Treatments that do not share the same letter are significantly different (alpha = 0.05, Tukey-adjusted). The error bars represent the standard error of the mean.

## Data Availability

The partial 16S rRNA gene sequences obtained in this study were deposited in GenBank under the accession numbers MW704083 through MW704095.

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
