# Peer review of "The Effects of Phyllosphere Bacteria on Plant Physiology and Growth of Soybean Infected with Pseudomonas syringae"

_plants, 2022, doi:10.3390/plants11192634_

Round 1

Reviewer 1 Report

Dear author,

Major:

- Must have been some kind of error in the formation of the PDF on pages 1 and 2, the title is on one page and the abstract on another.

- The Fig.1 in my PDF file has been cut I can only see the beginning of the phylogenetic tree.

Experimental overview

-In the methods I do not find what method was followed to count the colonies of the experiment in Fig.2. Above all, what was done to differentiate the Psg cells from the BC? and only count the Psg cells? this point needs to be clarified

­-Line 86: “We reported the  data for the day 35 only but the overall trend of differences across different treatments was similar on day 20 and 45 of plant growth (data not shown)”. Although as the authors said there was no difference in the overall trend in the treatments with respect to plus or minus BC, it would be interesting to see if the psg cells per leaf, plus/minus BC, increases or decrease between days 20 to 45 of cultivation. That is why I recommend that figure 2 be redone and the results of days 20 and 45 included and discussed.

The data for days 20 and 35 for the tables 1-3, could be included as supplemental material.

Minor:

-Line 103: “colony counter” manual or automatic counter? Please specified

-Line 153: “Approximately, 2x107 cells/mL were used to spray the soybean plants on day 16 and 31 of plant growth” It had already been mentioned previously in the line 115

This sentence, Line 170 “In effect, our goal ………………….with Psg. Line q 175. I think it would be better fix in the results or even discussion rather than as part of the methods

I understand that the BC was  sprayed before the psg infection. As it says in materials and methods. Line 114 “The BC was sprayed on the soybean plants after 12 and 24 days of plant growth and  Psg cells (2x107 cells/mL) were applied on day 16 and 31”. Therefore, I would rewrite this sentence, Line 346-“the BC applied to Psg infected plants ameliorates the negative effects of Psg”.  Since, this phrase seems to say that BC is applied but after psg infection.

-Line 375: “and BC added to infected plants”. Same case as mentioned above.

At the end of the discussion, a paragraph summarizing the main conclusions would be useful.

I encourage the authors to continue with this study. It would be interesting to know which of the 13 strains studied has a greater effect in protecting soybeans against infection.

Reviewer 2 Report

Phyllosphere bacteria are an important determinant of plant growth and resistance to pathogens. The authors observed a significant positive effect of the BC application in controlling Psg infection, as well as a positive influence of BC inoculation on the growth and physiological activities of the BC-treated plants, which looks very interesting. The experiments have been described in detail and the results are discussed adequately. The manuscript is well written. While generally well done, this reviewer thinks this MS could be accepted after major revision.

I have included below a number of concerns.

1.  I can’t see Figure 1 in current version.

2.  I suggest using bar figures to show the results of Table 1, 2 and 3 since it is a science paper.

3.   For the whole paper, the authors need to display some phenotype pictures to show the positive influence of BC inoculation on the growth and in controlling Psg infection, especially for figure 2.

Line 75-76: what is the exact growth condition for the soybean, like the temperature, light, and so on?

Line 82-83: re-write this sentence. It makes me confused.

Line 87-88: suggest to show the similar data in supplemental tables or figures rather than just say it.

Line 90: what is V4-5 growth-stage?

Line 321-323: “unlike for shoot mass”, Is it correct? I think the results were similar that the BC has positive effects on shoot mass and nodule mass.

Line 395: (41) should be [41]. Check all the similar mistakes, like 52, 80.

Line 408: “a previous study by [53] reported”, delete “by”.

Round 2

Reviewer 1 Report

Dear authors,

I believe that all my suggestions have been responded to favorably. I accept the paper in its current version.

Reviewer 2 Report

I believe the authors have done a great job addressing all my questions and concerns, so I think the revised manuscript is publishable in Plants. One more suggestion, the bar figures are too big, and the authors can make them more professionally.